# Association of Human Leukocyte Antigen Alleles with COVID-19 Severity and Mortality in a Spanish Population

**DOI:** 10.3390/medicina60091392

**Published:** 2024-08-25

**Authors:** Ester Lobato-Martinez, Javier Muriel-Serrano, Elena García-Payá, Pilar Gonzalez-de-la-Aleja, Raquel Garcia-Sevila, Mercedes Navarro-de-Miguel, Francisco Marco-de-la-Calle, Jose-Manuel Ramos-Rincon, Rosario Sanchez-Martinez

**Affiliations:** 1Internal Medicine Department, Dr. Balmis Universitary General Hospital, Avenida Pintor Baeza, 12, 03010 Alicante, Spain; 2Alicante Institute for Health and Biomedical Research (ISABIAL), Centro de Diagnóstico, Edif Gris, Planta 5ª, Avenida Pintor Baeza, 12, 03010 Alicante, Spain; 3Clinical Analysis Department, Dr. Balmis Universitary General Hospital, Avenida Pintor Baeza, 12, 03010 Alicante, Spain; 4Infectious Diseases Unit, Dr. Balmis Universitary General Hospital, Avenida Pintor Baeza, 12, 03010 Alicante, Spain; 5Pneumology Department, Dr. Balmis Universitary General Hospital, Avenida Pintor Baeza, 12, 03010 Alicante, Spain; 6Immunology Department, Dr. Balmis Universitary General Hospital, Avenida Pintor Baeza, 12, 03010 Alicante, Spain; 7Clinical Medicine Department, Miguel Hernández University, N-332, 87, 03550 Alicante, Spain

**Keywords:** SARS-CoV-2, COVID-19, genetic predisposition to disease, polymorphism, genetic, HLA antigens, hospitalization, mortality

## Abstract

*Background and Objectives*: The aim of the following cross-sectional study is to determine the association between human leukocyte antigen (HLA) alleles and outcomes in patients presenting to the emergency department (ED) with SARS-CoV-2 infection. *Methods and Materials*: Genotyping was made using the Axiom Human Genotyping SARS-CoV-2 Research Array. Statistical analysis was made with Fisher’s exact test and multivariable logistic regression, adjusted for sex, age and clinical variables. *Results*: Of 190 patients, 11.1% were discharged from the ED; 57.9% were admitted to the COVID-19 ward, without intensive care unit (ICU) admission; 15.3% survived an ICU admission; and 15.8% died. After multivariable analysis, two HLA alleles protected against hospital admission (HLA-C*05:01, adjusted odds ratio [aOR] 0.2, 95% confidence interval [CI] 0.055–0.731; and HLA-DQB1*02:02, aOR 0.046, CI 0.002–0.871) and one was associated with higher risk for ICU admission or death (HLA-DQA1*05:01, aOR 2.517, CI 1.086–5.833). *Conclusions*: In this population, HLA-C*05:01 and HLA-DQB1*02:02 are associated with a protective effect against hospital admission and HLA-DQA1*05:01 is associated with higher risk of ICU admission or death in the multivariable analysis. This may help stratify risk in COVID-19 patients.

## 1. Introduction

Human leukocyte antigen (HLA) genes play an important role in adaptive immune response [1], since they code the cell membrane proteins responsible for the antigen presentation to the T cells [2]. The HLA loci are located on the short arm of chromosome 6, which is the highest polymorphic region in the human genome [3]. The high polymorphism affects the peptide binding by HLA proteins, which in turn affects the host cellular and humoral immune response against different antigens [2], thus leading to a diverse spectrum of disease susceptibilities and severities [3]. The role of HLA genes as risk factors for severe clinical outcomes has been studied in many viral infections, such as hepatitis C virus (HCV), human immunodeficiency virus (HIV) [1], the coronaviruses responsible for severe acute respiratory syndrome (SARS) and Middle Eastern respiratory syndrome (MERS) [4].

Soon after the outbreak of the COVID-19 pandemic, reports began to emerge that infection rates, severity and mortality worldwide differed from country to country [3]. In Spain, from the beginning of the pandemic until June 2023, 681,927 people were admitted to hospital, 56,249 were admitted to the ICU and 121,852 died as a result of COVID-19 [5]. This disease has a broad clinical spectrum, presenting as anything from a mild, flu-like disease [3] to acute respiratory distress syndrome requiring intensive care and ventilation support [1]. Besides the classical risk factors linked to severe COVID-19 disease, such as age, sex and comorbidities, several studies have also found an association between polymorphisms in genes involved in the immune response, like HLA genes [6].

Some of the HLA association studies available in the literature have been performed in Spanish patients, either as a single population [7,8] or as part of a multicenter study [9,10,11,12]. While some of them have shown associations between HLA alleles and COVID-19 susceptibility, clinical severity or mortality [7,10,11,12], others did not find any statistically significant results between HLA alleles and COVID-19 clinical outcomes [8,9,13]. The disparity in results, paired with the scarcity of significant findings in HLA class II alleles (only the study from Lorente et al. [7] found a class II allele associated with higher COVID-19 mortality), highlight the need to perform more association studies in similar populations in order to compare the results, on the one hand, and to explore new associations, on the other hand.

Although HLA sequencing techniques are currently considered the gold standard, it is an expensive and technically complex method, which makes it challenging to use in large-scale studies [13]. Therefore, we used an imputation method of genetic variants typed with a genotyping array [14], which has a good concordance rate compared to the HLA sequencing of 0.939 in European Americans [15], which we assume to be similar to the concordance in European Spanish [16]. In order to impute the obtained results from the genotyping array, there are different imputation methods, such as the HLA*IMP:02 imputation program, which interrogates random nucleotides and uses linkage disequilibrium to infer the whole sequence from population databases. It achieves an average 4-digit imputation performance of 97% on European panels and a 2-digit accuracy over 90% for other ethnicities [14].

The AxiomTM Human Genotyping SARS-CoV-2 array [17] includes >800,000 single nucleotide polymorphisms (SNPs) of COVID-19 susceptibility, severity and immune response. It has already been used in other studies of genetic susceptibility in SARS-CoV-2 infection [18,19,20].

The aim of this study was to determine the association between HLA alleles and clinical severity and mortality in patients with active SARS-CoV-2 infection in a Spanish population.

## 2. Materials and Methods

### 2.1. Patients and Setting

This cross-sectional study took place from 3 March 2020 to 31 April 2021 at the Dr. Balmis General University Hospital (Alicante, Spain), located on the southern Mediterranean coast. Eligible patients were adults admitted to the hospital’s emergency department (ED) and diagnosed with COVID-19 pneumonia using the reverse transcriptase polymerase chain reaction (RT-PCR) test for SARS-CoV-2, who gave their informed consent to participate in the study and had a good quality blood sample to perform the genetic analysis.

### 2.2. Variables

Electronic medical records were reviewed to collect patient outcomes, which were grouped according to COVID-19 clinical severity as follows. Group A comprised patients with SARS-CoV-2 infection discharged from the ED without need for hospital admission (not admitted in hospital). Group B was made up of patients with severe infection admitted into the COVID-19 ward, who did not require intensive care unit (ICU) admission and were later discharged (admission and severe COVID-19). Group C included those admitted into the ICU with a very severe infection, who survived and were later discharged (ICU admission without in-hospital death [IHD]). Finally, Group D was composed of people who died during hospital admission (either on the COVID-19 ward or in the ICU) due to a very severe COVID-19 infection. The following clinical variables were also recorded:Sex: male or female.Age: at the moment of presentation in the ED.Hypertension: history of pharmacologically treated hypertension.Diabetes: history of type 1 or type 2 diabetes mellitus, under pharmacological treatment.Dementia: history of Alzheimer’s disease, Lewy bodies dementia, vascular dementia, frontotemporal dementia, or unspecified cognitive impairment with Global Disability Scale (GDS) score 4–7.Lung disease: history of asthma, chronic obstructive pulmonary disease, pulmonary hypertension, sleep apnea hypopnea syndrome, obesity hypoventilation syndrome, idiopathic pulmonary fibrosis or other types of chronic lung disease.Cardiovascular disease: history of congestive heart failure with preserved or reduced left ventricle ejection fraction, ischemic heart disease, moderate or severe mitral or aortic valve disease, miocardiopathy, periphery arterial disease or stroke. Atrial fibrillation with no other heart condition is not included.Kidney disease: history of chronic kidney disease from stage 3 to stage 5 (including hemodialysis and kidney transplant), chronic glomerular or tubular disease.Immunosuppression: history of primary or secondary immune deficiency. HIV infection with undetectable viral load and with active treatment when presenting at the ED is not considered as immunosuppression.Cancer: history of active cancer and/or under active treatment at the moment of presentation at the ED.Smoking history: either active, former, or no smoking history.Country: country of birth. Electronic clinical history does not record the patient’s race.

### 2.3. DNA Isolation and HLA Typing

The genomic DNA of all included patients was extracted from peripheral blood samples stored in our hospital biobank using the MagNA Pure Compact System (Roche Diagnostics GmbH, Mannheim, Germany). DNA purity was evaluated using OD260/OD280 and OD260/OD230 ratios (NanoDropTM One/One Microvolume UV-Vis Spectrophotometers, Thermo Fisher Scientific, Wilmington, DE, USA).

All participants were examined via whole-genome genotyping by Axiom Human Genotyping SARS-CoV-2 Research Array (Thermo Fisher Scientific, Waltham, MA, USA), following the manufacturer’s recommendations for the gene Titan multi-channel instrument. Arrays were analyzed using Axiom Analysis Suite 5.2 and Axiom HLA Analysis Software [17] using default parameters, and GRCh38 was used for genomic assembly. Both class I (HLA-A, B and C) and class II (HLA-DPA1, DPB1, DQA1, DQB1, DRB1, DRB3, DRB4 and DRB5) HLA alleles were analyzed.

Axiom Human Genotyping software uses the imputation model HLA*IMP:02 [14] and a reference panel of SNP array data from genome-wide association studies in multiple populations to statistically infer HLA types from human samples [17]. This method does not account for homozygosity and null alleles are not recognized by the software [17].

### 2.4. Statistical Analyses

All statistical analyses were performed using IBM SPSS Statistics for Windows, Version 26.0 (Armonk, NY, USA, IBM Corp.). For the descriptive analysis, categorical variables were expressed as frequencies (percentages) and continuous variables as medians (interquartile range, IQR). The outcomes used in the HLA allele analysis were hospital admission (group A [not admitted in hospital] versus groups B + C + D), ICU admission or IHD (group A + B versus group C + D) and IHD (group A + B + C versus group D). The differences in demographic characteristics were compared between the four groups independently (group A versus B versus C versus D), and also grouped by all three outcome analyses (group A versus group B, C and D; group A and B versus group C and D; and group A, B and C versus group D).

For every outcome and HLA allele, a contingency table was built, and allelic frequency differences were analyzed using Fisher’s exact test, with Bonferroni correction for multiple comparisons. Statistical significance was set at *p* < 0.05. For every separate HLA allele, all inconclusive or unavailable results were treated as missing data and excluded from the respective contingency table. We did not exclude the whole genetic information of all patients with any missing information so as not to lose valuable genetic information. Therefore, frequencies are calculated from different totals for every allele.

Multivariable analysis with a logistic binary regression model was performed on the alleles with statistically significant results in the univariable analysis, using sex, age and the clinical variables as covariates. Some alleles had a frequency of zero for some outcome groups, precluding multivariable analysis.

### 2.5. Ethical Aspects

This work was approved by the institutional research ethics committee of Dr. Balmis General University Hospital–ISABIAL (PI2020-089). All patients gave their informed consent before participating in the study. The research was conducted according to the principles of the Declaration of Helsinki.

## 3. Results

From the patients attending the ED with confirmed COVID-19 infection and who gave their informed consent to participate in the study, a total of 190 had a good quality peripheral blood sample and were included in the study. Table 1 shows the demographic characteristics of each group of the study population. As for the country of origin, 78% of the patients (148/190) were Spanish, 17% (32/190) were from South American countries with Hispanic heritage, 1% (2/190) were from other European countries and 4% (8/190) were from non-European, non-Hispanic countries.

### 3.1. Univariable Analysis

The analyses for all outcomes and allelic frequencies are presented in the Appendix A. HLA alleles showing statistically significant differences for any outcome in the univariable analysis are summarized in Table 2. In the crude analysis, four class I and seven class II HLA alleles showed statistically significant differences in at least one of the studied outcomes. After Bonferroni correction, significance held only for HLA-DRB4*01:03, showing a protective effect against the composite outcome of ICU admission and death.

### 3.2. Multivariable Analysis

Table 3 shows the results of the multivariable analysis adjusted for sex, age and clinical variables. Altogether, three HLA alleles showed independent and statistically significant relationships with COVID-19 outcomes: two had a protective effect against hospital admission (HLA-C*05:01nand HLA-DQB1*02:02), and one was associated with a higher risk of the composite outcome of ICU admission or death (HLA-DQA1*05:01). All these alleles have a population frequency higher than 0.1 in Spanish allelic databases [21].

In all cases, the effect was lost after applying Bonferroni correction, meaning that the observed associations may be influenced by the analysis of multiple comparisons. Multivariable analysis was not possible in seven cases in which the frequency of the HLA alleles in any of the outcome groups was zero.

## 4. Discussion and Limitations

### 4.1. Discussion

Since the beginning of the COVID-19 pandemic, different authors have studied the association between HLA alleles and the severity of the SARS-CoV-2 infection. When comparing our results with the published literature, HLA-C*05:01 was also related with COVID-19 outcomes in other studies, but with conflicting results. Poulton et al. [22] found an association with lower disease severity, coinciding with our analyses, whereas Sakuraba et al. [23] reported an association with a higher risk of COVID-19 death. Moreover, in another review [24], HLA-C*05:01 was associated with a higher risk of severe COVID-19 and death in multiple populations, including Spain. Other HLA alleles associated with COVID-19 susceptibility, severity or mortality in other studies including Spanish populations yielded no statistically significant associations in our study, such as HLA-A*03, HLA-B*39, HLA-C*16 and HLA-A*32 [7] for COVID-19 infection susceptibility; HLA-C*04:01 [10] and HLA-A*01 [12] for disease severity; and HLA-A*11, HLA-C*01, DQB1*04 [7] and HLA-A*02:01 [11] for death.

The HLA system is highly polymorphic, entailing important differences between geographical areas; for this reason, association studies in the literature often report different and sometimes conflicting results [4]. In this case, however, the different results can be explained, as Sakuraba et al. [23] include in their paper, by the interaction between HLA-C*05:01 and a type of killer cell immunoglobulin-like receptor, KIR2DS4fl, which recognizes peptides presented by HLA-C*05:01 and enhances immune activation, making patients more likely to suffer a cytokine storm. Therefore, the different results found in HLA-C*05:01 between populations, such as ours, may be linked to KIR2DS4fl allelic frequencies.

Our study is the first to show an association between two other HLA alleles and COVID-19 outcomes: HLA-DQB1*02:02, with a protective effect against hospital admission, and HLA-DQA1*05:01, associated with a higher risk of ICU admission or IHD. HLA-DQA1*05:01 was also associated with a higher risk of hospitalization in the univariable analysis, but it was not possible to adjust the analysis for covariates because one of the allelic frequencies was zero. Some studies [25] have shown a protective effect against COVID-19 of other HLA-DQ haplotypes, such as the well-known HLA-DQ2/8, which could be favored by an increased affinity of the HLA variant for SARS-CoV-2 peptide antigens, which in turn enhances the pathogen-immune system activation. The fact that other HLA-DQ polymorphisms show association with COVID-19 outcomes can shed more light into its role in SARS-CoV-2 infection.

Some of the alleles demonstrating a significant association with COVID-19 outcomes in this population are also associated with other viral diseases. For instance, HLA-C*05:01 expression may be downregulated in influenza infection [26]. Lastly, in a HLA expression study in patients with recurrent respiratory papillomatosis [27], the expression of HLA-DQB1*02:02 was selectively enriched, especially in patients with severe compared with moderate disease; with these results, the authors suggest that HLA-DQB1*02:02 may predict recurrent respiratory papillomatosis severity, but it is not related to a predisposition to the disease.

### 4.2. Limitations

The main strength of this study is that it is one of the few published studies on the topic in a Spanish population. However, it also has some limitations. First of all, despite the good concordance rate of HLA imputation programs compared to sequencing [15], this method can only genotype up to approximately 45% of the known HLA alleles, with Class I alleles being better covered than Class II [13]. Moreover, there may be errors in the reference panels due to failures in genotyping, which can affect imputation accuracy, and the average error rate is between 3 and 6% in high resolution (4-digit) HLA alleles [15]. It is also important to note that null alleles are not recognized by imputation methods. Despite these limitations, the AxiomTM Human Genotyping SARS-CoV-2 array [17] has already been used in other studies of genetic susceptibility in SARS-CoV-2 infection [18,19,20]. Moreover, Shen et al. [18] and Breno et al. [19] also study the association between HLA alleles and COVID-19 severity. All three articles also use imputation methods, and Breno et al. [18] used the HLA*IMP:02 imputation method, the one that we performed in our study.

When it comes to the statistical analysis, the modest sample size meant that many HLA alleles showed rather low frequencies, in many cases under 5 and even 0 for some alleles. Thus, it was not possible to adjust analyses for age and sex in all cases. This also means that the results obtained are imprecise, with wide confidence intervals and the disappearance of statistical significance after correcting for multiple comparisons. Missing data are also a limitation; when performing the genetic analysis, in some cases, we could not obtain all the alleles of an individual. This can be either because of the technical limitations, which we explain in a former response, because of problems with the quality of the sample, or both. The research team discussed what to do with the missing data, and we decided to not exclude the patients with missing alleles so as not to lose valuable genetic information. Moreover, the genetic analysis had missing data for some individuals, so the allelic frequencies are incomplete in some of the HLA groups.

## 5. Conclusions

The analysis of HLA alleles in a Spanish population identified three alleles that may influence the outcome of COVID-19 infection: two with a protective effect for hospital admission (HLA-C*05:01 and HLA-DQB1*02:02), and one associated with a higher risk for the combined outcome of ICU admission and IHD (HLA-DQA1*05:01). These findings, along with other genetic and demographic risk factors, may help to stratify the risk of patients with SARS-CoV-2 infection that seek hospital care, allowing to identify patients who are more likely to have complications due to COVID-19 early. It also provides insight into the immune system interaction in viral infections, which can help with other diseases.

## Figures and Tables

**Table 1 medicina-60-01392-t001:** Demographic characteristics of the sample, according to clinical outcome group.

	(A) Not Admitted in Hospital	(B) Admitted to COVID-19 Ward	(C) Admitted to ICU	(D) In-Hospital Death *	Total	*p* (A vs. B vs. C vs. D)	*p* (A vs. B + C + D)	*p* (A + B vs. C + D)	*p* (A + B + C vs. D)
**Gender, n (%)**									
Male	9 (8.6)	62 (59.6)	14 (13.5)	19 (18.3)	104	0.44	0.151	0.553	0.241
Female	12 (14)	48 (55.8)	15 (17.4)	11 (12.8)	86				
**Age, median (IQR)**	52 (35–63)	56 (45–71)	57 (51–72)	78 (72–88)	60 (47–74)	**<0.001**	**0.043**	**<0.001**	**<0.001**
**Hypertension, n (%)**									
Yes	5 (23.8)	37 (33.6)	9 (31)	20 (66.7)	71	**<0.001**	0.13	**0.019**	**<0.001**
No	16 (76.2)	73 (66.4)	20 (69)	10 (33.3)	119				
**Diabetes, n (%)**									
Yes	1 (4.8)	13 (11.8)	7 (24.1)	13 (43.3)	34	**<0.001**	0.077	**<0.001**	**<0.001**
No	20 (95.2)	97 (88.2)	22 (75.9)	17 (56.7)	156				
**Dementia, n (%)**									
Yes	0 (0)	4 (3.6)	0 (0)	3 (10)	7	**<0.001**	0.434	0.376	0.08
No	21 (100)	106 (96.4)	29 (100)	27 (90)	183				
**Lung disease, n (%)**									
Yes	4 (19)	18 (16.4)	2 (6.9)	15 (50)	39	**<0.001**	0.561	**0.046**	**<0.001**
No	17 (81)	92 (83.6)	27 (93.1)	15 (50)	151				
**Cardiovascular disease, n (%)**									
Yes	2 (9.5)	10 (9.1)	0 (0)	13 (43.3)	25	**<0.001**	0.456	**0.016**	**<0.001**
No	19 (90.5)	100 (90.9)	29 (100)	17 (56.7)	165				
**Kidney disease, n (%)**									
Yes	1 (4.8)	4 (3.6)	0 (0)	8 (26.7)	13	**<0.001**	0.566	**0.019**	**<0.001**
No	20 (95.2)	106 (96.4)	29 (100)	22 (73.3)	177				
**Immunosupression, n (%)**									
Yes	0 (0)	5 (4.5)	1 (3.4)	6 (20)	12	**<0.001**	0.234	**0.041**	**0.004**
No	21 (100)	105 (95.5)	28 (96.6)	24 (80)	178				
**Cancer, n (%)**									
Yes	0 (0)	4 (3.6)	2 (6.9)	8 (26.7)	14	**<0.001**	0.182	**0.002**	**<0.001**
No	21 (100)	106 (96.4)	27 (93.1)	22 (73.3)	176				
**Smoker status, n (%)**									
Active	2 (9.5)	5 (4.5)	2 (6.9)	3 (10)	12	**<0.001**	0.637	0.156	**0.001**
Former	7 (33.3)	25 (22.7)	5 (17.2)	16 (53.3)	53				
Never	12 (57.1)	80 (72.7)	22 (75.9)	11 (36.7)	125				
**Total (%)**	21 (11.1)	110 (57.9)	29 (15.3)	30 (15.8)	190				

* In-hospital death: Died during hospitalization (in conventional ward or in ICU).

**Table 2 medicina-60-01392-t002:** HLA class I and II allele frequencies with statistically significant differences in the study population in the univariable analysis.

HLA Allele	Hospital Admission				ICU Admission/Death				Death			
	Yes (B, C, D) n (%)	No (A) n (%)	*p*	Corrected * *p*	Yes (C, D) n (%)	No (A, B) n (%)	*p*	Corrected * *p*	Yes (D) n (%)	No (A, B, C) n (%)	*p*	Corrected * *p*
B*07:05	0/144 (0)	2/19 (10.05)	0.013	0.48	0/50 (0)	2/113 (1.8)	1	ns	0/26 (0)	2/137 (1.5)	1	ns
B*14:01	3/144 (2.1)	0/19 (0)	1	ns	3/50 (6)	0/113 (0)	0.028	1	2/26 (7.7)	1/137 (0.7)	0.067	ns
B*35:03	3/144 (2.1)	0/19 (0)	1	ns	3/50 (6)	0/112 (0)	0.028	1	3/26 (11.5)	0/137 (0)	0.004	0.15
C*05:01	13/167 (7.8)	5/21 (23.8)	0.035	0.70	5/58 (8.6)	13/130 (10)	1	ns	2/30 (6.7)	16/158 (10.1)	0.74	ns
DPA1*02:02	3/167 (1.8)	0/21 (0)	1	ns	3/59 (5.1)	0/129 (0)	0.030	0.15	2/30 (6.7)	1/158 (0.6)	0.067	ns
DQA1*01:01	23/162 (14.2)	7/20 (35)	0.027	0.19	8/54 (14.8)	22/128 (17.2)	0.83	ns	5/28 (18.5)	25/147 (16.1)	0.78	ns
DQA1*05:01	41/162 (25.3)	0/20 (0)	0.008	0.056	19/54 (35.2)	22/128 (17.2)	0.011	0.077	7/28 (25.9)	34/147 (21.9)	0.62	ns
DQB1*02:02	1/155 (0.6)	2/20 (10)	0.035	0.49	0/53 (0)	3/122 (2.5)	0.55	ns	0/28 (0)	3/147 (2)	1	ns
DRB1*01:01	5/142 (3.5)	1/18 (5.6)	0.52	ns	3/48 (6.3)	3/112 (2.7)	0.37	ns	3/23 (13)	3/137 (2.2)	0.039	1
DRB1*15:01	12/142 (8.5)	1/18 (5.6)	1	ns	8/48 (16.7)	5/112 (4.5)	0.022	0.73	4/23 (17.4)	9/137 (6.6)	0.096	ns
DRB4*01:03	15/163 (9.2)	5/20 (25)	0.049	0.15	1/58 (1.7)	19/125 (15.2)	0.005	0.015	0/30 (0)	20/153 (13.1)	0.048	0.14

* Bonferroni correction. ICU: intensive care unit; ns: non-significant. d.

**Table 3 medicina-60-01392-t003:** Adjusted odds ratio for HLA alleles with statistically significant differences in the univariable analysis.

**Hospital Admission (B, C and D vs. A)**			
**HLA Allele**	** *p* **	**a*p***	**aOR (95%CI)**
**B*07:05**	0.013	N/C	N/C
**C*05:01**	0.035	**0.015**	**0.2 (0.055–0.731)**
**DQA1*01:01**	0.027	0.121	0.408 (0.131–1.266)
**DQA1*05:01**	0.008	N/C	N/C
**DQB1*02:02**	0.035	**0.040**	**0.046 (0.002–0.871)**
**DRB4*01:03**	0.049	0.165	0.408 (0.115–1.446)
**ICU admission/Death (C and D vs. A and B)**			
**HLA allele**	** *p* **	**a*p***	**aOR (95%CI)**
**B*14:01**	0.028	N/C	N/C
**B*35:03**	0.028	N/C	N/C
**DPA1*02:02**	0.03	N/C	N/C
**DQA1*05:01**	0.011	**0.031**	**2.517 (1.086–5.833)**
**DRB1*15:01**	0.022	0.069	3.643 (0.905–14.662)
**DRB4*01:03**	0.005	0.069	0.140 (0.017–1.164)
**Death (D vs. A, B and C)**			
**HLA allele**	** *p* **	**a*p***	**aOR (95%CI)**
**B*35:03**	0.004	N/C	N/C
**DRB1*01:01**	0.039	0.110	197.275 (0.304–128221.91)
**DRB4*01:03**	0.048	N/C	N/C

***p***: unadjusted p. a***p***: adjusted p. aOR: adjusted odds ratio. 95 CI%: confidence interval. Significant *p* is highlighted in bold. N/C: not calculated (in the cases any of the frequencies equals zero).

## Data Availability

The raw data supporting the conclusions of this article will be made available by the corresponding author on request.

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
