# Peer review of "Association of Human Leukocyte Antigen Alleles with COVID-19 Severity and Mortality in a Spanish Population"

_medicina, 2024, doi:10.3390/medicina60091392_

Round 1

Reviewer 1 Report

Comments and Suggestions for Authors

see attached

Author Response

Dear reviewer,

Thank you for your comments and suggestions. The changes are highlighted in the revised manuscript. However, we detail our response below:

Title: Authors are recommended to avoid abbreviations in the title.

Thank you for your recommendation, we have changed the title to avoid abbreviations.

Abstract: introduce this acronym for the first time.

Thank you for your comment, we have introduced the acronym in the abstract.

Introduction: since the authors have used the virus names abbreviation in the last two examples, it would be better to be consistent and use abbreviations for all examples

Thank you for your suggestion, we have used abbreviations for all examples in the manuscript in order to improve consistency.

Introduction: I found these two paragraphs to be irrelevant to the introduction section. Moreover, the importance of this study and the gap in the knowledge that the current study is aiming to address is extremely lacking. the introduction section not only sets the context but also should clearly highlights the significance of the study and the gap in the existing knowledge that the current research is trying to address.

Thank you for your comments regarding the introduction section. We have added a paragraph in the Introduction section (lines 60 – 69) in which we highlight the existing gaps in the previously published material in the topic that we intend to address with our study, with 6 new references (7 – 12). We have also linked this new introductory section with our findings in the Discussion section (lines 235 – 240). After discussing it among the research team, we decided to keep the paragraphs about the array in the introduction because we would like to set the context of the method we used to the readers.

Methods: sample size justification is missing

Thank you for pointing it out. We included patients attended in the Emergency Department with confirmed COVID-19 infection, who gave their consent to participate in the study and who had a good quality blood sample. After the recruitment period ended, we had 190 valid patients, which constitute our sample. We have added further explanation in the Methods (lines 96-98) and Results section (lines 180-182).

Results: Authors are recommended to highlights the major findings instead of duplicating data already presented in tables

Thank you for your recommendation, we have eliminated the numerical data from the text.

Table 2: ???

We apologise for the confusion it may have caused; we wrote it since the table heading and the table itself were in different pages inside the manuscript. We have taken it out.

Discussion: put this paragraph under the heading "limitation"

Thank you for your suggestion, we have added the heading “Limitations” to the manuscript, including the paragraph mentioned.

Reviewer 2 Report

Comments and Suggestions for Authors

General Comments:
This manuscript describes association of HLA alleles with COVID-19 severity and mortality in a Spanish population. The study was conducted in one site, Dr. Balmis General University Hospital located on the southern Mediterranean coast in Spain.

The analysis resulted in finding two new HLA alleles potentially relate to COVID-19 outcomes. HLA-DQB1*02:02, had a protective effect against hospital admission, and HLA-DQA1*05:01, associated with a higher risk of ICU admission or IHD.

The main limitation of the study is the sable size, limiting detailed subgroup analysis, as stated by authors.

Specific comment:

Line 144: Clarification is warranted if demographic characteristics (table 1) were tested using the structure (group A versus groups B + C + D), (group A + B versus group C + D), (group A + B + C versus group D), or (group A vs B vs C vs D).

Author Response

Dear reviewer,

Thank you for your comments and suggestions. The changes are highlighted in the revised manuscript. However, we detail our response below:

Line 144: Clarification is warranted if demographic characteristics (table 1) were tested using the structure (group A versus groups B + C + D), (group A + B versus group C + D), (group A + B + C versus group D), or (group A vs B vs C vs D).

Thank you for pointing it out. The p value we included originally in Table 1 was for the comparison group A vs B vs C vs D. Since we have p values for all comparisons, we have added them to Table 1. We have also specified it in the Methods section (lines 159-162).
